# Morphogenesis and morphometry of brain folding patterns across species

Sifan Yin[1], Chunzi Liu[1], Gary PT Choi[2], Yeonsu Jung[1], Katja Heuer[3], Roberto Toro[3], L Mahadevan[1,4,5]*

[1]School of Engineering and Applied Sciences, Harvard University, Cambridge, United States; [2]Department of Mathematics, The Chinese University of Hong Kong, Hong Kong, Hong Kong; [3]Institut Pasteur, Université Paris Cité, Unité de Neuroanatomie Appliquée et Théorique, Paris, France; [4]Department of Physics, Harvard University, Cambridge, United States; [5]Department of Organismic and Evolutionary Biology, Harvard University, Cambridge, United States

## eLife Assessment

This **important** study presents a cross-species and cross-disciplinary analysis of cortical folding. The authors use a combination of physical gel models, computational simulations, and morphometric analysis, extending prior work in human brain development to macaques and ferrets. The findings support the hypothesis that mechanical forces driven by differential growth can account for major aspects of gyrification. The evidence presented is overall strong and **convincingly** supports the central claims; the findings will be of broad interest in developmental neuroscience.

*For correspondence:
lmahadev@g.harvard.edu

**Abstract** Evolutionary adaptations associated with the formation of a folded cortex in many mammalian brains are thought to be a critical specialization associated with higher cognitive function. The dramatic surface expansion and highly convoluted folding of the cortex during early development is a theme with variations that suggest the need for a comparative study of cortical gyrification. Here, we use a combination of physical experiments using gels, computational morphogenesis, and geometric morphometrics to study the folding of brains across different species. Starting with magnetic resonance images of brains of a newborn ferret, a fetal macaque, and a fetal human, we construct two-layer physical gel brain models that swell superficially in a solvent, leading to folding patterns similar to those seen in vivo. We then adopt a three-dimensional continuum model based on differential growth to simulate cortical folding in silico. Finally, we deploy a comparative morphometric analysis of the in vivo, in vitro, and in silico surface buckling patterns across species. Our study shows that a simple mechanical instability driven by differential growth suffices to explain cortical folding and suggests that variations in the tangential growth and different initial geometries are sufficient to explain the differences in cortical folding across species.

## Introduction

Although not all brains are folded, in many mammals, the folded cerebral cortex is known to be critically important for brain cognitive performance and highly dependent on the hierarchical structure of its morphology, cytoarchitecture, and connectivity (*Gautam et al., 2015*; *Suárez et al., 2020*; *Pang et al., 2023*). Brain function is thus related both to the topological structure of neural networks (*Bullmore and Bassett, 2011*), as well as the geometry and morphology of the convoluted cortex (*Kriegeskorte and Wei, 2021*), both of which serve to enable and constrain neuronal dynamics (*Pang et al., 2023*). Across species, cortical morphologies show a large diversity, as shown in *Figure 1a*

**eLife digest** The most recognizable feature of the human brain is its surface folding patterns. In humans and many other mammals, the outer layer of the brain – the cerebral cortex – develops a complex pattern of ridges and grooves. These folds allow a large cortical surface to fit inside the skull and are closely linked to brain function.

Cortical folding can begin before birth, but it continues after birth as the gray matter cortex grows faster than the softer tissue beneath it, known as white matter. When a growing surface is constrained in this way, mechanical stresses build up and cause it to buckle and form sharp creases or sulci like those in the palm of one's hand. Although genes control how brain cells grow and move, physical forces determine how this growth is translated into shape through mechanical instabilities. Different species, such as ferrets, macaques and humans, show distinct folding patterns, raising the question of whether these differences require specific biological mechanisms or can arise from a shared physical process.

Yin et al. asked whether brain folding across different mammalian species can be explained by the same basic physical mechanism: differential growth of the cortex relative to the underlying tissue. While a related study (Choi et al.) showed that this mechanism can reproduce folding in the human brain, it was unclear whether it could also account for the diversity of folding patterns seen across species. Resolving this question helps clarify how universal physical principles interact with biological growth during brain development.

The results show that a single mechanical mechanism can explain brain folding in ferrets, rhesus macaques and humans. Yin et al. combined physical experiments using soft gel models shaped like fetal brains that swell and fold when immersed in a solvent, computer simulations of growing brain tissue, and quantitative comparisons of folding patterns using mathematical frameworks. In all species studied, faster growth of the cortical layer caused the surface to buckle and form realistic folds. Differences between species were explained by variations in initial brain shape, and cortical growth rate, rather than by different folding rules. The close match between real brains, gel models, and simulations supports differential growth as a unifying explanation for cortical folding.

The work of Yin et al. provides a simple physical framework for understanding how brain folding arises across species. This suggests that genetic control of brain shape and cortical expansion could, over time, lead to different patterns of brain folding across species, as well as disorders associated with abnormal folding. However, before these insights can inform medical or evolutionary applications, future studies will need to link specific genes and cellular processes to the growth parameters used in the models and test whether the framework can explain individual differences and disease-related folding patterns, the subject of a companion paper in the same journal.

(*Takahata et al., 2012*; *Heuer et al., 2019*). And within our own species and in model organisms, such as the ferret used to study the genetic precursors of misfolding, cortical folding, and misfolding are known to be markers of healthy and pathological neurodevelopment, disease, and aging (*Hutton et al., 2009*; *de Moraes et al., 2024*) (see also *Appendix 1—figure 1*; *Oegema et al., 2020*; *Choi et al., 2025*). Thus, a comparative study of cortical folding is essential for understanding brain morphogenesis and functionalization across evolution (*Pang et al., 2023*; *Schwartz et al., 2023*), during development as well as in pathological situations associated with disease.

The development of cortical morphology involves the coordinated and localized expression of many genes that lead to the migration and differentiation of neural stem and progenitor cells (*van der Meer and Kaufmann, 2022*; *Oegema et al., 2020*; *Del-Valle-Anton and Borrell, 2022*). All these biological processes cooperatively generate an expansion of the cortex relative to the underlying white matter and eventually drive cortical folding (*Akula et al., 2023*). While a range of mechanisms have been proposed in the past for the processes leading to folding (*Striedter et al., 2015*; *Holland et al., 2015*; *Van Essen, 2020*), over the past decade, theoretical and experimental evidence have converged on the primary determinant of folding as a mechanical instability associated with the formation of a localized crease or sulcus (*Toro and Burnod, 2005*; *Hohlfeld and Mahadevan, 2012*; *Tallinen et al., 2013*) driven by differential growth, with iterations and variations that qualitatively explain the brain gyrification (*Tallinen et al., 2014*; *Tallinen et al., 2016*). However, combining this

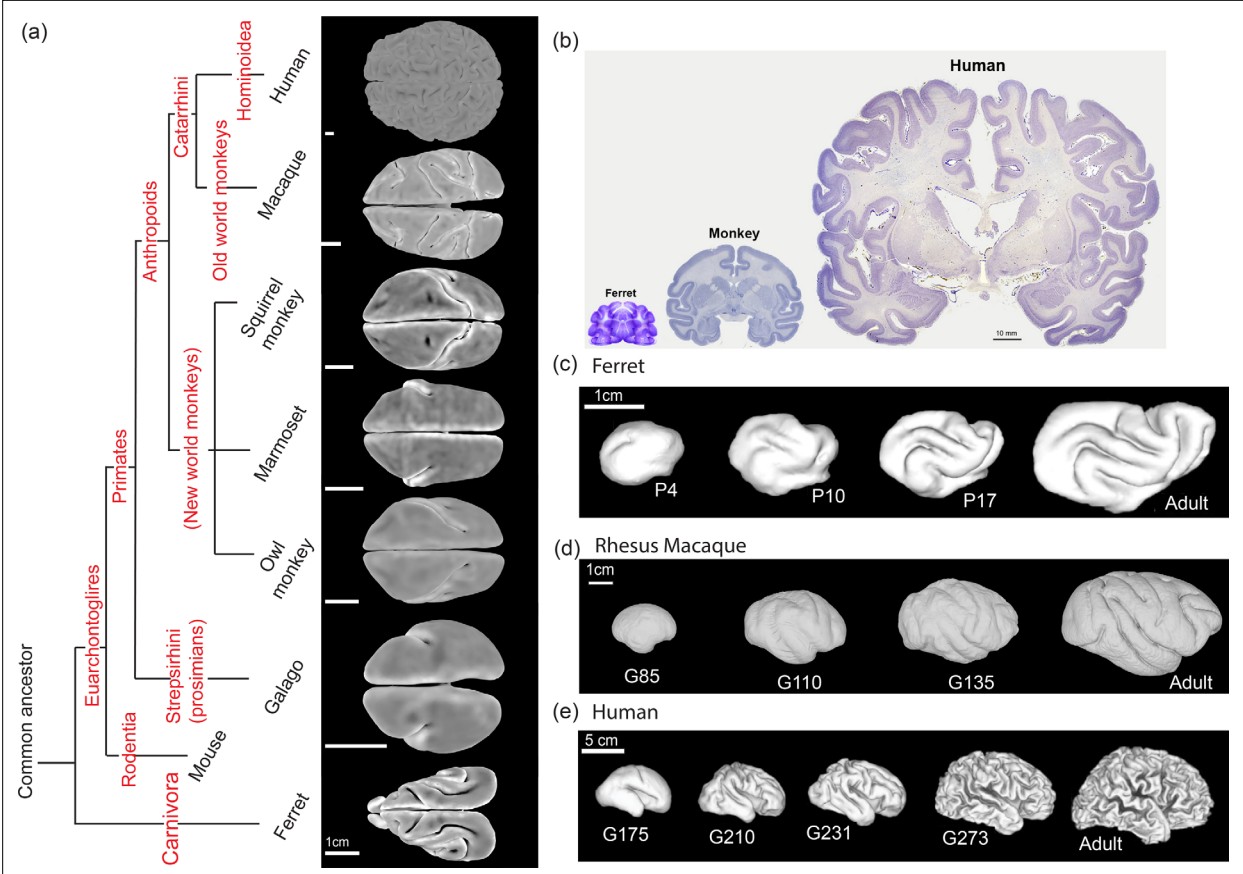

**Figure 1.** The diversity of the cortical morphologies and developmental processes across species. (**a**) Phylogenetic relationship of species. Adapted from *Takahata et al., 2012*; *Heuer et al., 2019*. Typical real brain surfaces of ferrets and primates are presented. Color represents mean curvature. Scale bars: 1 cm (estimated from *Herculano-Houzel, 2009*). (**b**) Stained sections of mature brain tissue from ferret, rhesus macaque, and human. Scale bar: 10 mm. Adapted from *Noctor, 2016*. (**c–e**) 3D reconstruction of cortical surfaces of ferret, macaque, and human brains from fetal to adult. (**c**) Ferret: postnatal day 4, 10, 17, and adult maturation (*Barnette et al., 2009*). Scale bar: 1 cm; (**d**) Macaque: gestation day 85, 110, 135 (*Liu et al., 2020*), and adult maturation (*Calabrese et al., 2015*). Scale bar: 1 cm; (**e**) Human: gestation day 175 (week 25), 210 (week 30), 231 (week 33), 273 (week 39), and adult maturation (*Barnette et al., 2009*). Scale bar: 5 cm.

mechanistic model with a comparative perspective that aims to quantify the variability of folding patterns across species, or linking it to genetic perturbations that change the relative expansion of the cortex remain open questions. In a companion paper (*Choi et al., 2025*), we address the latter using the ferret as a model organism, while in the current study, we address the former question using a combination of physical experiments with gel swelling, numerical simulations of differential growth and geometric morphometrics to compare brain morphogenesis in the ferret, the macaque, and the human. The species were selected as they have very different brain sizes and folding patterns (*Figure 1b*) and thus represent different branches in the evolutionary tree, Carnivora, Old World monkeys, and Hominoidea. Furthermore, in all three species, we have access to a time course of the development of the folds, as shown in *Figure 1c*.

## Results

### Experiments on swelling gel-brains

To mimic the mechanical basis for brain morphogenesis based on the differential growth of the cortex relative to the white matter, we used the swelling of physical gels that mimic the fetal brain developmental process during post-gestation stages. Previously, this principle has been demonstrated

**Video 1.** Part 1: The swelling processes of physical gel brains of ferret, macaque, and human brains; Part 2: The simulations mimicking the developmental processes of fetal brains of ferret, macaque, and human from smooth surface to the convoluted pattern.

https://elifesciences.org/articles/107138/figures#video1

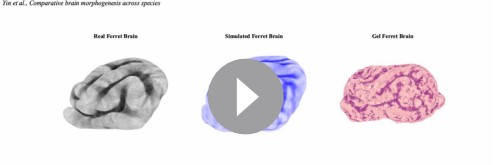

**Video 2.** Comparison of the real (gray), gel (pink), and simulated (blue) ferret brains. Color represents shape index.

https://elifesciences.org/articles/107138/figures#video2

for *Homo sapiens* (human) brain development (*Tallinen et al., 2014*; *Tallinen et al., 2016*). To demonstrate that the same principle applies to other species, we constructed physical gels from the fetal brain MRI scans for *Macaca mulatta* (rhesus macaque) and *Mustela furo* (ferret). In brief, a two-layer PDMS gel is constructed from the 3D fetal brain MRI reconstruction and immersed in an organic solvent. Immersion causes solvent imbibition into the surface of the physical gel, which swells leading to a compressive strain in the outer layers that causes the surface layer to form convolutions that resemble the folding patterns in the brain cortex layer. Time-lapse images of the gel model to mimic brain folding in *Macaca mulatta* (rhesus macaque) up to G110 are shown in *Figure 2a*, while in *Figure 2b*, we show the initial and final states of swelling to mimic different post-gestation stages corresponding to G85, G110, and G135. A visual inspection of the swollen gels constructed from different post-gestation stages showed qualitatively different folding patterns, indicating the sensitivity of the folds to the initial undulations present on the physical gel surfaces. In *Figure 2(c)* and *Video 1*, we show the results of similar physical gel experiments to mimic brain folding morphogenesis for *Homo sapiens* (*Tallinen et al., 2016*) and *Mustela furo Choi et al., 2025*; in each case, the initial condition was based on 3D fetal brain MRIs and the final state was determined qualitatively using the overall volume of the brain relative to the initial state. No attempt was made to vary the swelling ratio of the surface as a function of location, although it is likely that in the different species this was not a constant. To quantitatively describe the folding patterns, the swollen gel surfaces were then scanned and reconstructed using X-ray Computed Tomography (Methods). *Videos 2–4* show the fetal brain MRI scans and the reconstructed 3D swollen gel surfaces in juxtaposition for *H. sapiens*, *M. furo*, and *M. mulatta*. This paves the way for a quantitative comparison of the results of the physical experiments with those derived from a mechanical theory for brain morphogenesis and those from scans of macaque, ferret, and human brains.

## Simulations of growing brains

To test the capability of the mechanical model for brain morphogenesis based on differential growth (*Tallinen et al., 2014*; *Budday et al., 2014*; *Tallinen et al., 2016*) to explain patterns across species, we perform numerical simulations of the developing brains of ferret, macaque, and human modeled as soft tissues. Here, we only consider the simplest homogeneous growth profile which is sufficient to capture the folding formation across different species; regional growth of the cortical layer based on real data from tracking the surface expansion of fetal brains (*Garcia et al., 2018*; *Alenyà et al., 2022*; *Weickenmeier, 2023*) is a more sophisticated alternative that we do not adopt for reasons of simplicity.

The initial brain models are reconstructed from 3D fetal brain MRI (*Methods*) and assumed to be composed of gray and white matter layers which are considered as hyperelastic materials with differential tangential growth ratio $g$. A multiplicative decomposition of deformation gradient gives $\mathbf{F} = \mathbf{A} \cdot \mathbf{G}$ with $\mathbf{A}$ the elastic part and $\mathbf{G} = \sqrt{g}\mathbf{I} + (1 - \sqrt{g})\mathbf{n} \otimes \mathbf{n}$ the growth tensor. We adopt a modestly compressible neo-Hookean material with strain energy density

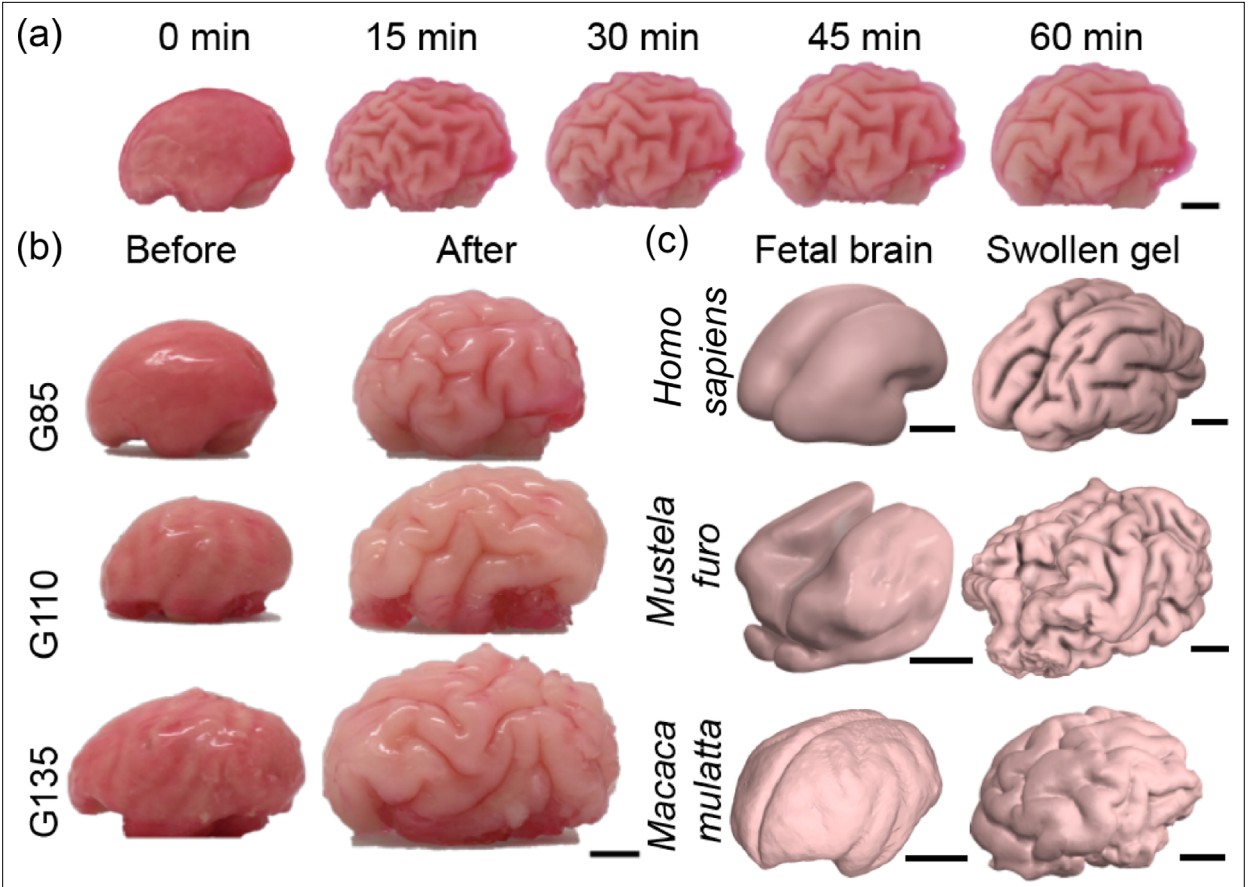

**Figure 2.** Physical gel model that recapitulates the growth-driven morphogenesis mechanism across phylogeny and developmental stages. (**a**) A time-lapse of the physical gel brain mimicking macaque brain development starting from G110. (**b**) Left views of three physical gels mimicking macaque post-gestation day 85, day 110, and day 135 before and after hexane swelling. Scale bar: 1 cm. (**c**) Comparison of fetal/newborn brain scans and the reconstructed surfaces of swollen physical gels for various species. Scale bars: 1 cm.

$$W = \frac{\mu}{2} \left[ J_{\mathrm{A}}^{-2/3} \mathrm{tr}(\mathbf{A}^{\mathrm{T}} \cdot \mathbf{A} - 3) \right] + \frac{K}{2} \left( J_{\mathrm{A}} - 1 \right)^2, \tag{1}$$

where $\mathbf{F}$ is the deformation gradient, $J_{\mathrm{A}} = \det\mathbf{A}$, $\mu$ is the initial shear modulus, and $K$ is the bulk modulus. Considering a modestly compressible material, we assume $K = 5\mu$.

We solve the final shapes of the growing tissues using a custom finite element method (*Tallinen et al., 2016*). All the initial geometries of smooth fetal brains are obtained from open data sources and through collaboration (*Methods*, 3D model reconstruction), and the growth ratio distribution is assumed as a function of the initial location, including the distance to the cortical surface and the presumably non-growing regions. Other parameters, such as the thickness ratio and modulus ratio of

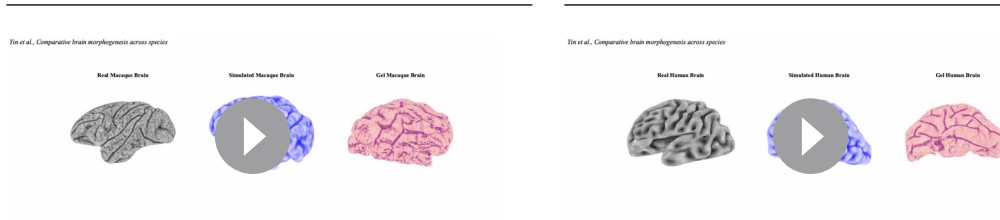

**Video 3.** Comparison of the real (gray), gel (pink), and simulated (blue) macaque brains. Color represents shape index.
https://elifesciences.org/articles/107138/figures#video3

**Video 4.** Comparison of the real (gray), gel (pink), and simulated (blue) human brains. Color represents shape index.
https://elifesciences.org/articles/107138/figures#video4

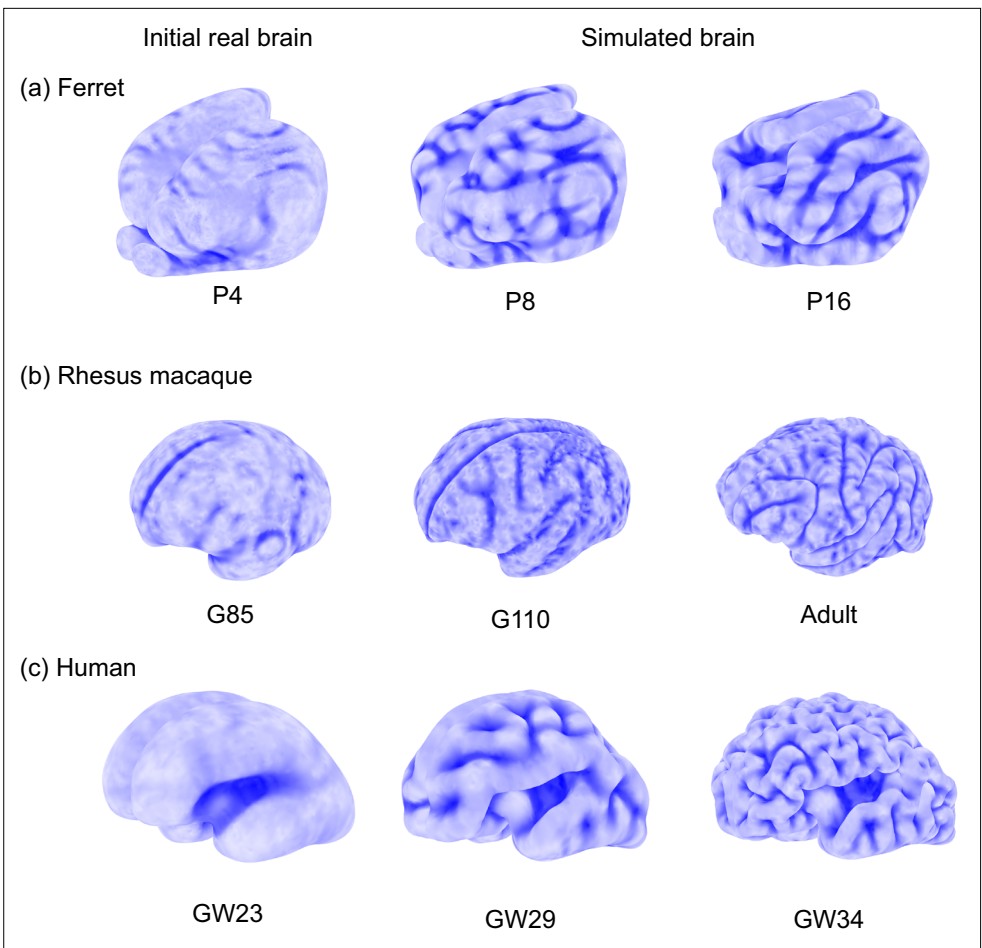

**Figure 3.** Simulations of growing brains of (**a**) ferret, (**b**) rhesus macaque, and (**c**) human. Starting from smooth fetal/newborn brains, simulations show different gyrification patterns across species. The brains are modeled as soft elastic solids with tangential growth in the gray matter (see *Simulations of growing brains* for details). Initial 3D geometries are taken from the reconstruction of MRI (see *Methods*, 3D model reconstruction). Mechanical parameters of growth ratio and cortical thickness are provided in *Table 2*. Color from dark to light blue represents shape index (as defined in *Equation 2*) from −1 to 1.

gray and white matter, and the temporal changes of growth ratios are assumed (*Methods*, numerical simulations, Table 2). The brain models are discretized to tetrahedrons by Netgen (*Schöberl, 1997*). An explicit algorithm is adopted to minimize the total strain energy of the deforming tissues. We adopt a step-wise simulation strategy where the initial geometry of each presumed stress-free state is obtained from real MRI data of earlier-stage fetal brains, instead of using a continuous model where only the initial smooth brain geometry is input (*Tallinen et al., 2016*). The simulated developmental processes of fetal brains are presented in *Figure 3* and *Video 1*.

## Morphometric analysis

To verify whether our simulation methods and physical gel models are sufficient to capture the developmental process of fetal brains and reproduce cortical patterns in adult brains, we compare the cortical surfaces of real, simulated, and gel brains across different species. *Figure 4a* presents the hemispherical cortical surfaces of the real, simulated, and gel brains (denoted as $\mathcal{S}_1$, $\mathcal{S}_2$, and $\mathcal{S}_3$, respectively). Left and right symmetry and the comparison of whole brains are presented in *Appendix 1—figure 3* and *Figure 4*, *Videos 2–4*. Major sulci are extracted and highlighted by hand for further alignment. To analyze the shape differences, we then adapted the parameterization methods in *Choi et al., 2020*; *Choi et al., 2015a*; *Choi and Lui, 2015b*; to map the brain surfaces onto a common disk shape with the major sulci aligned using landmark-matching disk quasi-conformal parameterizations. Denote

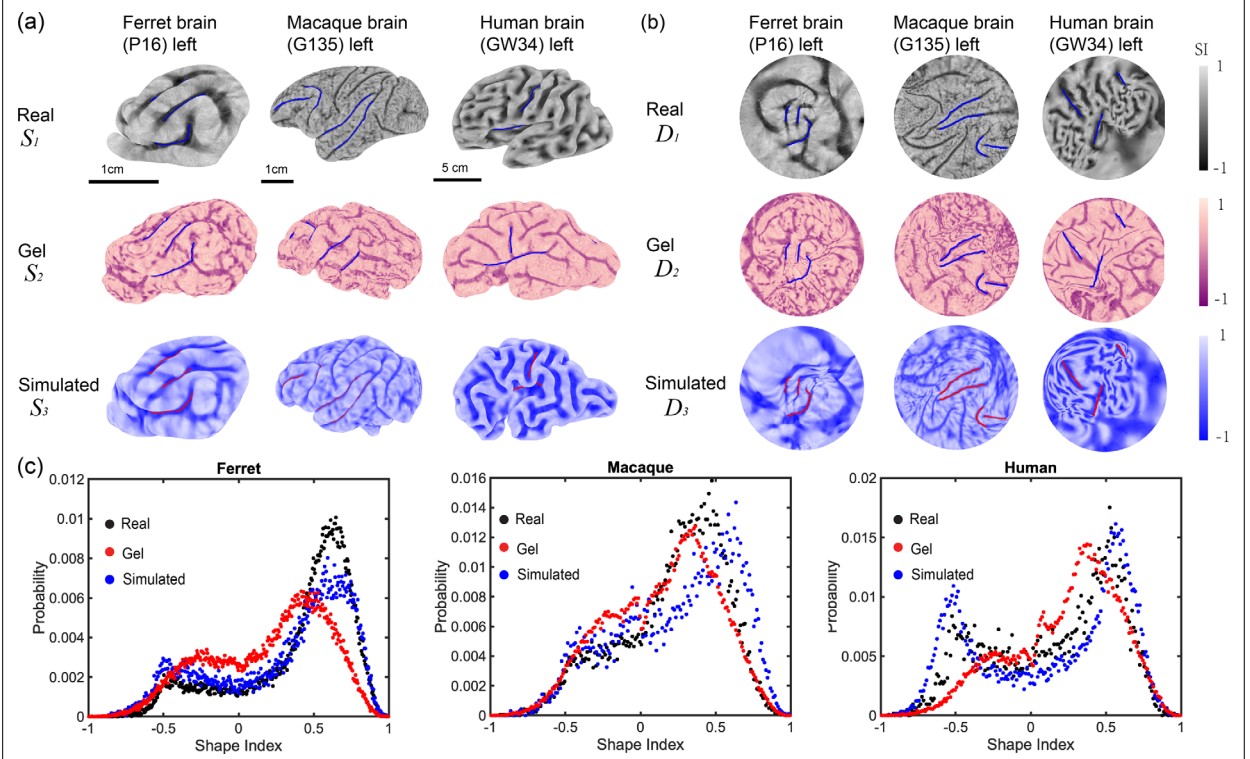

**Figure 4.** Comparison among real ($\mathcal{S}_1$), simulated ($\mathcal{S}_2$), and gel brains ($\mathcal{S}_3$) of ferret, rhesus macaque, and human via morphometric analysis. (**a**) 3D cortical surfaces of in vivo, in vitro, and in silico models. Left brain surfaces are provided here. The symmetry of the left and right halves of the brain surfaces is discussed in *Videos 2–4*, *Appendix 1—figures 3 and 4*. Three or four major folds of each brain model are highlighted and serve as landmarks. The occipital pole region of macaque brains remains smooth in real and simulated brains. (**b**) The quasi-conformal disk mapping with landmark matching of cortical surfaces on disk (see Sec *Morphometric analysis* for details). Blue or red curves represent corresponding landmarks. Color represents shape index (SI, as defined in *Equation 2*). Similarity indices of each simulated and gel brain surfaces are presented in *Table 1*. (**c**) Histogram of shape index of ferret, macaque, and human. Black, red, and blue dots represent the probability of shape index of real, gel, and simulated surfaces, respectively.

the three-disk parameterization results as $\mathcal{D}_1, \mathcal{D}_2, \mathcal{D}_3$, as shown in *Figure 4b*. Multiple quantitative measures, such as surface area, cortical thickness, curvature, and sulcal depth can be adopted to analyze curved surfaces and to compare different surfaces. Here, we use shape index (SI) and rescaled mean curvature $\tilde{H}$ to describe the similarity among real, simulated, and gel surfaces. Shape index is a dimensionless and scale-independent surface measure (*Koenderink and van Doorn, 1992*). It can be calculated from the mean and Gaussian curvatures by

$$\mathrm{SI} = \frac{2}{\pi}\arctan\left(\frac{H}{\sqrt{H^2 - K}}\right). \tag{2}$$

Shape index ranging within [−1, 1] defines a continuous shape change from convex, saddle, to concave shapes (see *Appendix 1—figure 2* for nine categories of typical curved surfaces). For brain surfaces, we can classify sulcal pits ($-1 < \mathrm{SI} < -0.5$), sulcal saddles ($-0.5 < \mathrm{SI} < 0$), saddles($\mathrm{SI} = 0$), gyral saddles ($0 < \mathrm{SI} < 0.5$) and gyral nodes ($0.5 < \mathrm{SI} < 1$). When the shape index equals −1 or 1, it represents a defect of the curvature tensor with two eigenvalues identical, as shown in *Appendix 1—figure 3*. The probability of shape index distribution exhibits two peaks, as shown in *Figure 4c*, corresponding to ridge and rut shapes (SI=±0.5), where the ridge shape (SI=0.5) is dominating. In contrast, the rescaled mean curvature histogram exhibits a unique peak around 0.2 (*Appendix 1—figure 3*). The two-peak and unique-peak distributions of adult human brain surfaces have also been presented in previous research (*Demirci and Holland, 2022*; *Hu et al., 2013*). To quantify the similarities between every two brain surfaces, we evaluate the distribution of $I(v)$ differences on the common

**Table 1.** Similarity index evaluated by comparing the shape index of simulated brains (S), swollen gel brain simulacrums (G), and real brain surfaces (R), calculated with vector p-norm $p = 2$, as described in *Equation 4*.

| Similarity Index | Simulation-reaGl | Gel-real | Simulation-gel |
|---|---|---|---|
| Ferret (left) | 0.7632 | 0.7307 | 0.7117 |
| Ferret (right) | 0.7339 | 0.7222 | 0.7019 |
| Macaque (left) | 0.7624 | 0.7608 | 0.7512 |
| Macaque (right) | 0.7559 | 0.7632 | 0.7463 |
| Human (left) | 0.6947 | 0.7425 | 0.7000 |
| Human (right) | 0.6909 | 0.7325 | 0.7001 |

disk domain at each vertex $v$. Here $I(v)$ represents either the surface shape index $\mathrm{SI}(v)$ or the rescaled mean curvature $\tilde{H}(v)$ defined as

$$\tilde{H}(v) = 2\frac{H(v) - \min(H)}{\max(H) - \min(H)} - 1. \tag{3}$$

Thus, the similarity $s$ of the distributions $I_{\mathcal{S}_1}, I_{\mathcal{S}_2}$ of the two surfaces $\mathcal{S}_1, \mathcal{S}_2$ is then evaluated via

$$s(\mathcal{S}_1, \mathcal{S}_2) = 1 - \frac{1}{2m^{1/p}} \left\| I_{\mathcal{S}_1} - I_{\mathcal{S}_2}(g_2^{-1} \circ f_{12} \circ g_1) \right\|_p, \tag{4}$$

where $m$ is the total number of vertices, $g_1 : \mathcal{S}_1 \to \mathcal{D}_1$ and $g_2 : \mathcal{S}_2 \to \mathcal{D}_2$ are the initial disk conformal parameterizations, $f_{12}$ is the landmark-aligned quasi-conformal map between $\mathcal{D}_1$ and $\mathcal{D}_2$, and $\| \cdot \|_p$ is the vector p-norm:

$$\left\| I_{\mathcal{S}_1} - I_{\mathcal{S}_2}(g_2^{-1} \circ f_{12} \circ g_1) \right\|_p = \left( \sum_{k=1}^{m} \left| I_{\mathcal{S}_1}(v_k) - I_{\mathcal{S}_2}((g_2^{-1} \circ f_{12} \circ g_1)(v_k)) \right|^p \right)^{1/p}. \tag{5}$$

Note that since $I_{\mathcal{S}_1} \in [-1, 1]$ and $I_{\mathcal{S}_2} \in [-1, 1]$, we have $0 \leq \left\| I_{\mathcal{S}_1} - I_{\mathcal{S}_2}(g_2^{-1} \circ f_{12} \circ g_1) \right\|_p \leq (2^p m)^{1/p} = 2m^{1/p}$. Therefore, we always have $0 \leq s(\mathcal{S}_1, \mathcal{S}_2) \leq 1$.

We calculate the similarity indices with different p-norm: $p = 1$, $p = 2$, and $p = \infty$ of both rescaled mean curvature and shape index. The results are given in *Table 1*, *Appendix 1—table 2*.

## Discussion

In this study, we have explored the mechanisms underlying brain morphogenesis for a few different mammalian species. Using fetal and adult brain MRIs for ferrets, macaques, and humans, we carried out physical experiments using swelling gels, combined with a mathematical framework to model the differential growth of the cortex. Our study confirms that iterations and variations of a mechanical (sulcification) instability suffices to recapitulate the basic morphological development of folds. We then deployed a range of morphometric tools to compare the results of our physical experiments and simulations with 3D scans of real brains and show that our approaches are qualitatively and quantitatively consistent with experimental observations of brain morphologies.

All together, our study shows that differential growth between the gray matter cortex and white matter bulk provides a minimal physical model to explain the variations in the cortical folding patterns seen in multiple species. More specifically, we see that the overall morphologies are controlled by the relative size of the brain (compared to the cortex), as well as the scaled surface expansion rate, both of which can and do have multiple genetic antecedents (*Bae et al., 2014*; *Shinmyo et al., 2017*; *Qi et al., 2023*; *Lai et al., 2003*; *Barresi et al., 2024*; see *Appendix 1—table 1*).

Our results point to some open questions. First, the relationship between physical processes that shape organs and the molecular and cellular processes underlying growth has been the subject of many recent studies, e.g., in the context of gut development (*Gill et al., 2024a*; *Gill et al., 2024b*), and it would seem natural to expect similar relationships in brain development. There is a growing literature linking genes with brain malformation and pathologies. For example, GPR56 (*Bae et al.,*

**Table 2.** Parameters for numerical simulations.

| Species | Model | Growth ratio | Stiffness ratio | Normalized cortical thickness |
|---------|-------|--------------|-----------------|-------------------------------|
| Ferret | step-wise | 1.8 | 1 | $0.1-0.005t$ |
| Macaque | step-wise | 1.8 | 1 | $0.1-0.05t$, $0.1-0.1t$ |
| Human | continuous | 1.8 | 1 | $0.05-0.03t$ |

2014) and Cdk5 (*Shinmyo et al., 2017*) can affect progenitors and neurons in migration, SP0535 (*Qi et al., 2023*) can affect neural proliferation, and foxp2 can affect neural differentiation (*Lai et al., 2003*; *Barresi et al., 2024*), all of which change the cortical expansion rate and thickness, consequently leading to brain malformation and pathologies, as listed in *Appendix 1—table 1*. While a direct relation between gene expression levels and the effective tangential growth rate $\mathbf{G}$ and cortical thickness has only been partially resolved, as for example in our companion study on the folding and misfolding of the ferret brain (*Choi et al., 2025*), further studies are needed to address how genetic programs drive cell proliferation, migration, size, and shape change that ultimately lead to different cortical morphologies. Second, although our focus has been exclusively on the morphology of the brain, recent studies (*Pang et al., 2023*; *Schwartz et al., 2023*) are suggestive of a link between cortical geometry and function from both developmental and evolutionary perspectives and suggest natural questions for future study. Third, despite prescribing a simple spatially homogeneous form for the cortical expansion for fetal brain surfaces of all three species studied, we were able to capture the essential features of the folds and variations therein. The effect of spatio-temporally varying inhomogeneous growth needs to be further investigated by incorporating regional growth of the gray and white matter not only in human brains (*Garcia et al., 2018*; *Weickenmeier, 2023*) based on public datasets (*Namburete et al., 2023*), but also in other species to investigate folding differences across species, inter-individual variability, and finally regional differences in folding. More accurate and specific work is expected to focus on these directions. Finally, our physical and computational models, along with our morphometric approaches are a promising avenue to pursue in the context of the inverse growth problem, i.e., postulate the fetal brain morphologies from the adult brains. This may one day soon allow us to reconstruct the adult brain geometries from fossil endocasts (*de Sousa et al., 2023*), and eventually provide insights into how a few mutations might have triggered the rapid expansion of the cortex across evolutionary time and led to the convoluted human brain able to ponder how it might itself have folded.

# Materials and methods
## 3D model reconstruction
### Pre-processing
We used a publicly available database for all our 3D reconstructions: fetal macaque brain surfaces are obtained from *Liu et al., 2020* (https://www.nitrc.org/projects/fetalmacaatlas); newborn ferret brain surfaces are obtained from project FIIND *Toro et al., 2018*; and fetal human brain surfaces are obtained from *Tallinen et al., 2016*. These 2D manifolds of brain surfaces were first normalized by their characteristic lengths $L_0 = L_x^0/2$ and discretized to triangle meshes by Meshlab (*Cignoni et al., 2008*) and then converted to 3D models and discretized to tetrahedral elements in Netgen https://ngsolve.org/.

### Post-processing
All the numerically calculated brain surfaces and scanned gel brain surfaces were extracted as 3D triangle meshes. These intact cortical surfaces were then dissected to left and right semi-brains and normalized by half of their longitudinal lengths $L = L_x/2$. These surfaces were then checked and fixed to a simply connected open surface to satisfy the requirements for open disk conformal mapping (*Choi and Lui, 2015b*).

## Experimental protocol for gel experiments

The physical gel was constructed following a previous publication (*Tallinen et al., 2016*). In brief, a negative rubber mold was generated with Ecoflex 00–30 from a 3D-printed fetal brain plastic model. The core gel was then generated with SYLGARD 184 at a 1:45 crosslinker: base ratio. Three layers of PDMS gel at a 1:35 crosslinker:base ratio were surface-coated onto the core layer to mimic the cortex layer. Pigments were added to the PDMS mixture for bright-field visualization. To mimic the cortex folding process, the physical gel was immersed in n-hexane. The time-lapse videos were taken with an iPod Touch 7th Gen. To reconstruct the swollen gel surface and analyze the folding patterns, swollen gels were imaged with a Nikon X-Tek HMXST X-ray CT machine. The voxel resolution for all scans was 100 μm. To minimize solvent evaporation during the 30 min scan, cotton soaked in the solvent was placed inside the container. The container's thin acrylic walls allowed for clear X-ray transmission. The container's thin acrylic walls allowed for clear X-ray transmission. To test the reversibility of the folding pattern formation, the physical gel models were allowed to dry in a laminar flow hood overnight before being immersed in hexane again.

## 3D reconstruction of swollen gel surface from X-ray CT

The z-stack images obtained from the X-ray CT machine were segmented by a machine-learning-based segmentation toolbox, Labkit, via ImageJ (*Schneider et al., 2012*). A classification was created for each swollen gel. Then, a 3D surface of the segmented gel image was generated by ImageJ 3D viewer. Further post-processing was conducted in SOLIDWORKS and MeshLab (*Cignoni et al., 2008*), including reversing facial normal direction, re-meshing, and surface-preserving Poisson smoothening.

## Numerical simulations

The simulation geometries of ferret, macaque, and human are based on MRI of smooth fetal or newborn brains. For the ferret, we take the P0, P4, P8, and P16 postnatal brains as initial shapes of the step-wise growing model; for the macaque, we take G80 and G110 fetal brains as initial shapes of the step-wise growing model. To focus on fold formation, we did not consider the lack of patterning in the relatively smooth regions, such as the Occipital Pole of the macaque; for the human, we take the GW22 fetal brain as the initial shape of the continuous growing model. Small perturbations of the initial geometry typically affect only the minor folds, while the main features of the major folds, such as their orientation, width, and depth, are well conserved across individuals (*Bohi et al., 2019*; *Wang et al., 2021*). For simplicity, we do not perturb the fetal brain geometry obtained from datasets. Both gray and white matter are considered as neo-Hookean hyperelastic material with the shear modulus distribution

$$\mu(d) = \mu_{\mathrm{w}} + \frac{\mu_{\mathrm{g}} - \mu_{\mathrm{w}}}{1 + \exp[10(d/h - 1)]},$$ 
(6)

where $d$ is the distance from an arbitrary material point inside the brain to the cortical surface, and $h$ is the approximated cortical thickness assumed to decrease with growth process time $t$ (*Tallinen et al., 2016*). The subscripts 'g' and 'w' represent gray and white matter, respectively. The tangential growth ratio $g$ has a similar spatial distribution

$$g(d) = g_{\mathrm{w}} + \frac{g_{\mathrm{g}} - g_{\mathrm{w}}}{1 + \exp[10(d/h - 1)]},$$ 
(7)

where $g_{\mathrm{w}}$ and $g_{\mathrm{g}}$ represent the growth ratio at cortical surface and at innermost white matter. The parameters used in simulations are listed in *Table 2*.

# Acknowledgements

We acknowledge partial financial support from the NSF-ANR grant 2204058 (LM, KH, RT), the Simons Foundation and the Henri Seydoux Fund (LM), the CUHK Faculty of Science Direct Grant for Research #4053650 (GPTC), the project NeuroWebLab (ANR-19-DATA-0025, KH, RT), DMOBE (ANR-21-CE45-0016), the European Union's Horizon 2020 research and Marie Skłodowska-Curie grant agreement 101033485 (KH).

## Additional information

### Funding

| Funder | Grant reference number | Author |
|---|---|---|
| NeuroWebLab | ANR-19-DATA-0025 | Katja Heuer<br>Roberto Toro |
| Agence Nationale de la Recherche | ANR-21-CE45-0016 (DMOBE) | Katja Heuer |
| European Union's Horizon 2020 | 101033485 | Katja Heuer |
| Agence Nationale de la Recherche | 2204058 | Katja Heuer<br>Roberto Toro<br>L Mahadevan |
| U.S. National Science Foundation | 2204058 | Katja Heuer<br>Roberto Toro<br>L Mahadevan |
| Simons Foundation | | L Mahadevan |
| Henri Seydoux Fund | | L Mahadevan |
| CUHK Faculty of Science Direct Grant for Research | 4053650 | Gary PT Choi |

The funders had no role in study design, data collection and interpretation, or the decision to submit the work for publication.

### Author contributions

Sifan Yin, Chunzi Liu, Investigation, Methodology, Writing – original draft, Writing – review and editing; Gary PT Choi, Methodology, Writing – original draft, Writing – review and editing; Yeonsu Jung, Methodology; Katja Heuer, Roberto Toro, Data curation; L Mahadevan, Conceptualization, Writing – original draft, Project administration, Writing – review and editing

### Author ORCIDs

Sifan Yin (ID) http://orcid.org/0000-0002-0296-3981
Gary PT Choi (ID) http://orcid.org/0000-0001-5407-9111
Katja Heuer (ID) https://orcid.org/0000-0002-7237-0196
L Mahadevan (ID) https://orcid.org/0000-0002-5114-0519

Reviewer #1 (Public review): https://doi.org/10.7554/eLife.107138.3.sa1
Reviewer #2 (Public review): https://doi.org/10.7554/eLife.107138.3.sa2
Author response https://doi.org/10.7554/eLife.107138.3.sa3

## Additional files

### Supplementary files

MDAR checklist

### Data availability

Reconstructed 3D surface models of fetal and adult brains of macaque and human are available on GitHub at https://github.com/YinSifan0204/Comparative-brain-morphologies (copy archived at *Yin, 2025*). Requests for the newborn ferret brain data should be made to *Toro et al., 2018* by contacting Roberto Toro (rto@pasteur.fr) or Katja Heuer (katjaqheuer@gmail.com). All other data are included in the article and/or appendix.

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

## Appendix 1

### Malformations of human cortical development

In the main text, we have shown examples of normal cortical development and the resulting folding processes by following fetal and adult animals across a few different species. Defects in neuronal migration disorders may lead to a group of rare brain malformations, the most severe forms being lissencephaly and polymicrogyria, as shown in *Appendix 1—figure 1a*; *Oegema et al., 2020*. Lissencephaly, or agyria-pachygyria, is characterized by a simplified convolutional pattern where a few broad gyri are separated by rudimentary primary fissures and sulci. Lissencephaly is accompanied by a very thick cortical gray matter layer (*Appendix 1—figure 1a*, middle) which has been confirmed as a critical cause in this malformation (*Budday et al., 2014*; *Tallinen et al., 2014*). Polymicrogyria, on the contrary, is an overly convoluted cortex with a reduced cortical thickness and an increasing number of secondary folds (*Appendix 1—figure 1a*, right). Various malformations of cortical development are congenital and genetically heterogeneous diseases in which mutations or deletions of genes have been identified. For example, a regional deletion mutation in a regulatory element of GP56 can selectively disrupt the human cortex surrounding the Sylvian fissure bilaterally, including 'Broca's area.' *Appendix 1—figure 1b* shows the MRI of polymicrogyria of a non-coding mutation in the GPR56 gene (*Bae et al., 2014*). This abnormally thin cortex is folded giving a paradoxical but characteristic thickened appearance. Many other genes have also been found associated with brain malformation and pathologies, such as foxp2 (*Lai et al., 2003*; *Barresi et al., 2024*), SP0535 (*Qi et al., 2023*) and Cdk5 (*Shinmyo et al., 2017*). These genes affect progenitors and neurons in migration, proliferation, and differentiation, which change the cortical expansion rate and thickness, consequently leading to brain malformation and pathologies, as listed in *Appendix 1—table 1*.

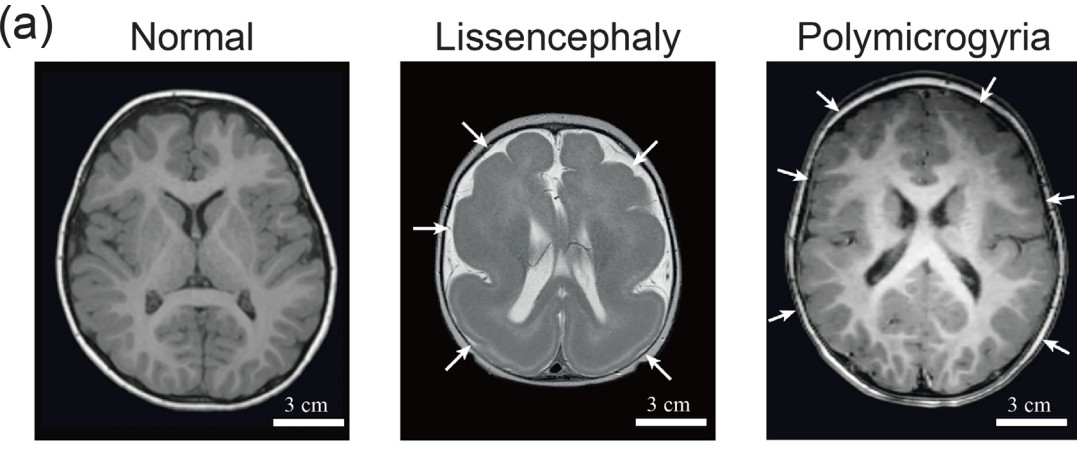

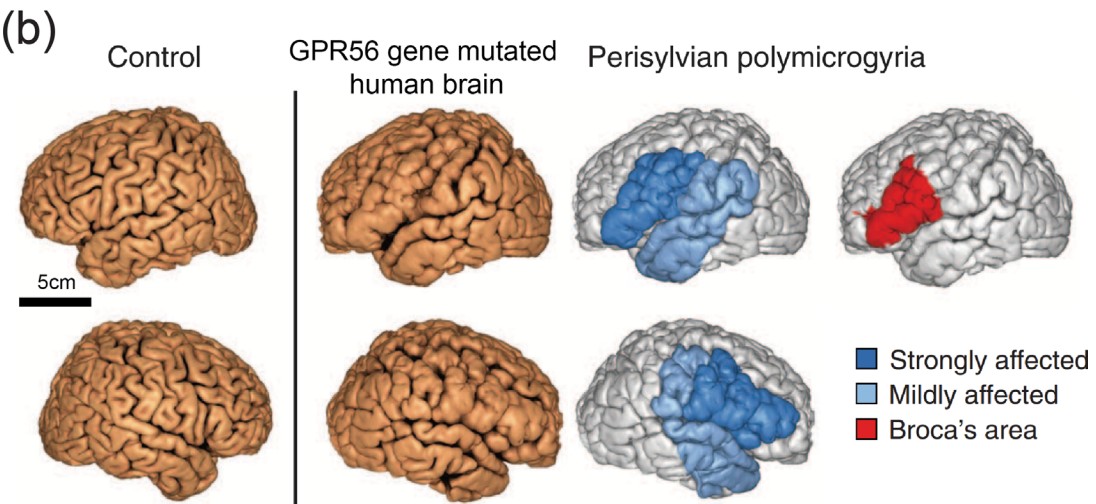

**Appendix 1—figure 1.** Human brain malformation. (**a**) MRI scans showing common malformations of cortical development of human brains. Adapted from *Oegema et al., 2020* with permission. Left: normal brain. Middle: lissencephaly spectrum with agyria–severe pachygyria (arrows). Right: bilateral frontoparietal polymicrogyria with abnormally small gyri and shallow sulci (arrows). Scale bars: 3 cm (estimated from *Mochida, 2009*). (**b**) A non-coding mutation in the GPR56 gene disrupts perisylvian gyri. MRI shows polymicrogyria in the perisylvian area, resulting in a characteristic, thickened appearance. Adapted from *Bae et al., 2014* with permission.

## Statistical characteristics of brain folding

To characterize the form of the convolutions on the surface of the brain, we use a dimensionless and scale-independent surface measure, the shape index (SI), to quantify 3D cortical morphologies across species. The definition of shape index is given as (*Koenderink and van Doorn, 1992*)

$$\text{SI} = \frac{2}{\pi} \arctan\left(\frac{H}{\sqrt{H^2 - K}}\right), \tag{8}$$

or equivalently,

$$\text{SI} = \frac{2}{\pi} \arctan\left(\frac{k_{\min} + k_{\max}}{k_{\min} - k_{\max}}\right), \tag{9}$$

where $k_{\max}$ and $k_{\min}$ are the maximum and minimum curvatures, respectively, $k_{\max,\min} = H \pm \sqrt{H^2 - K}$. The shape index is a continuous number representing the local shape of a surface. *Appendix 1— figure 2* illustrates a shape index scale divided into nine categories: spherical cup, trough rut, saddle

rut, saddle, saddle ridge, ridge, dome, and spherical cap, with the color representing the shape index value ranging from −1 to 1.

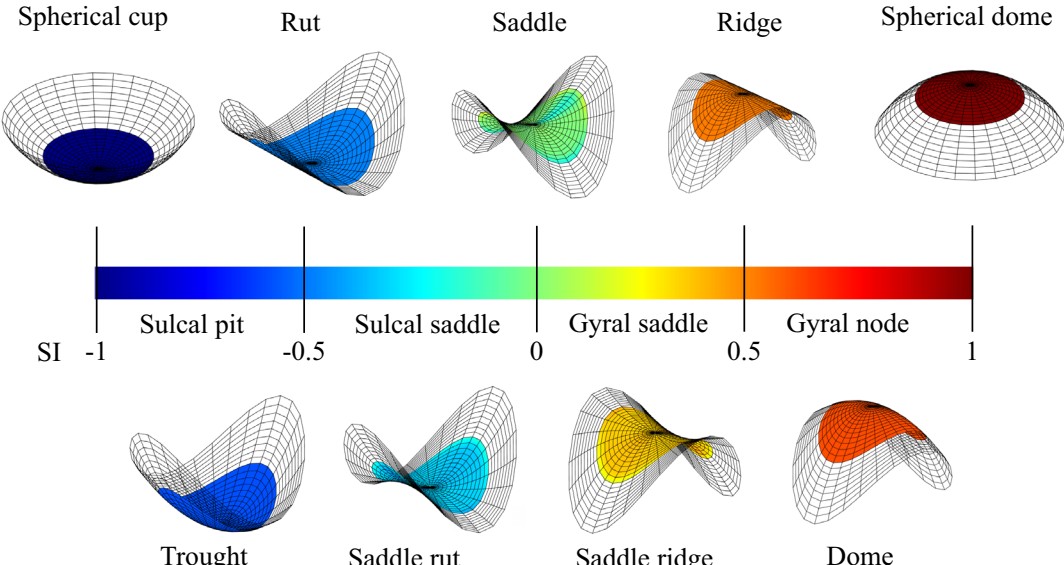

**Appendix 1—figure 2.** Illustration of shape index scale divided into nine categories: spherical cup, trough rut, saddle rut, saddle, saddle ridge, ridge, dome, and spherical cap. The insets are schematics of local curved surfaces. All outward normals pointing upwards.

In the main text, we have shown the shape index distribution of left semi-sphere brains. Here, we present the results of both left and right semi-spheres of ferret, macaque, and human brains in *Appendix 1—figure 3* (top two rows) and *Appendix 1—figure 4* with comparison among the real (black dots), gel (red dots), and simulated (blue dots) surfaces. The probability of shape index distribution exhibits two peaks, corresponding to ridge and rut shapes (SI=±0.5), where the ridge shape (SI=0.5) dominates. In contrast, the rescaled mean curvature histogram exhibits a unique peak around 0.2 (*Appendix 1—figure 3*, the third and fourth rows). These histograms illustrate that shape index and mean curvature exhibit qualitatively different distributions and both of these should be taken into account when describing cortical surface characteristics, consistent with earlier studies (*Demirci and Holland, 2022*; *Hu et al., 2013*).

**Appendix 1—table 1.** Gene-related brain properties and malformation.

| Gene | Change in geometry or physical properties | Malformations and dysfunctions |
|---|---|---|
| foxp2 *Barresi et al., 2024*; *Lai et al., 2003* | Reduced gray matter density (thinner cortex) in Broca's area | Polymicrogyria, speech, and language disorder |
| SP0535 *Qi et al., 2023* | Increased expansion of neural progenitors | improved cognitive ability and working memory |
| Cdk5 *Magen et al., 2015*; *Shinmyo et al., 2017* | Abnormalities in neuronal migration and organization | Lissencephaly with cerebellar hypoplasia (LCH) |
| GPR56 *Bae et al., 2014*; *Luo et al., 2011* | Neural progenitors migration | Bilateral frontoparietal polymicrogyria |
| PAFAH1B1, RELN, TUB1A1, DCX, ARX, NDE1 *Koenig et al., 2021*; *Verrotti et al., 2010* | Increased cortical thickness | Lissencephaly |

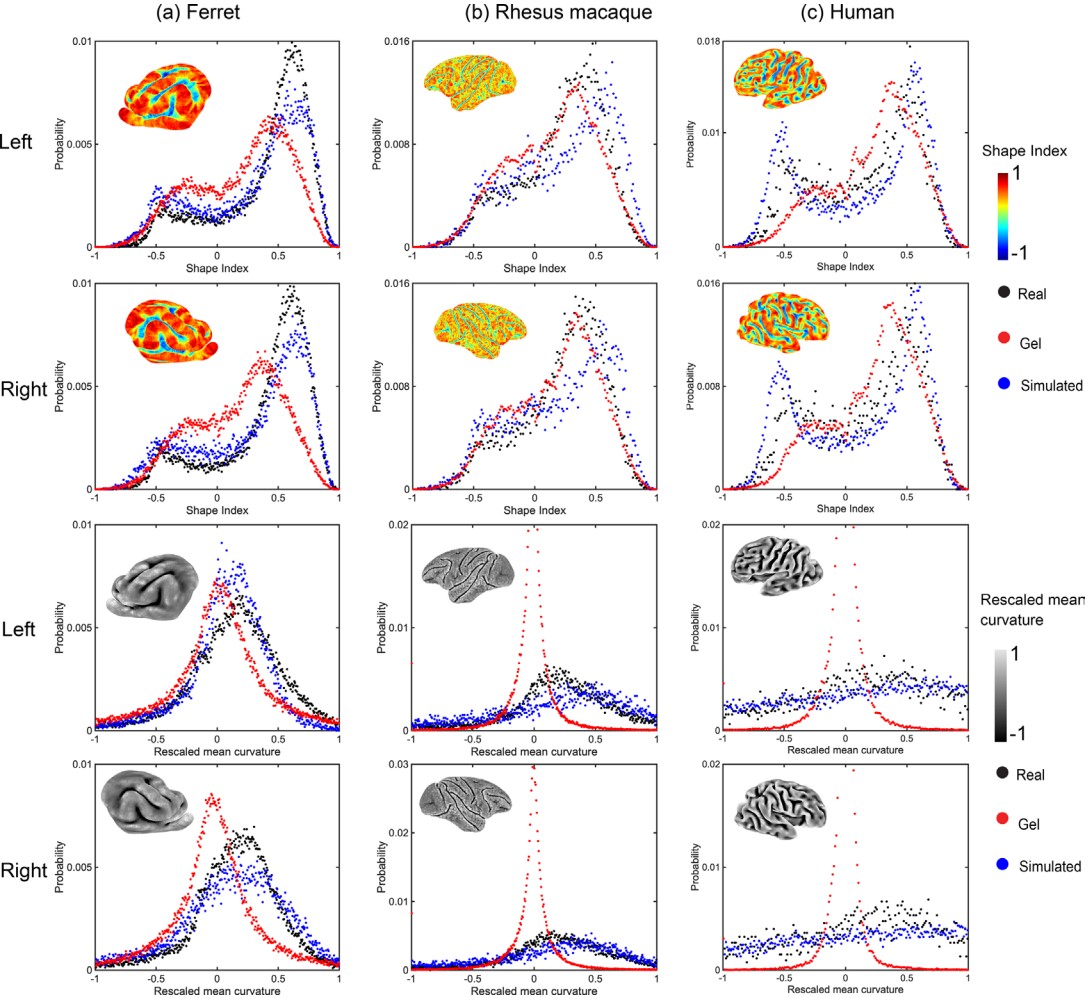

**Appendix 1—figure 3.** The histogram of shape index SI (top two rows) and rescaled mean curvature $\tilde{H}$ (bottom two rows) of adult cortical surfaces of ferret, macaque and human. Insets are real brain surfaces. Colors represent shape index SI (*Equation 2* in the main text) or rescaled mean curvature $\tilde{H}$ (*Equation 3* in the main text).

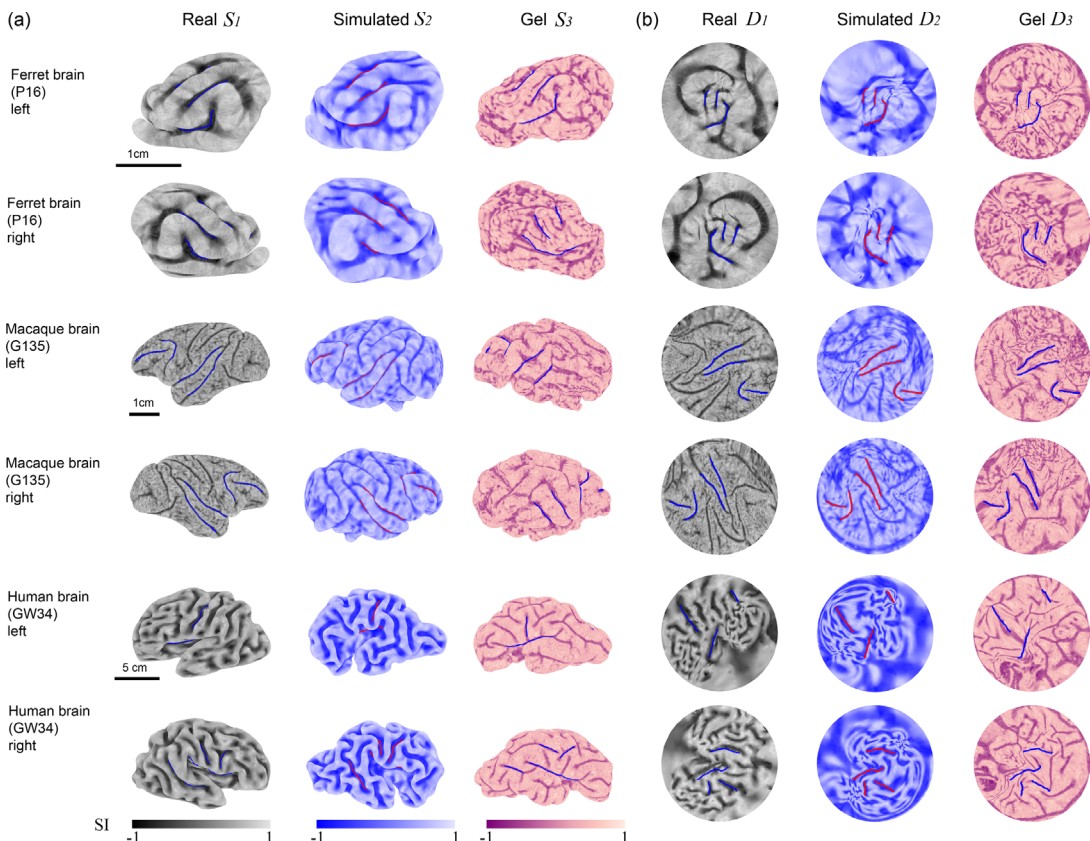

**Appendix 1—figure 4.** Comparison among real ($\mathcal{S}_1$), simulated ($\mathcal{S}_2$), and gel brains ($\mathcal{S}_3$) of ferret, rhesus macaque, and human via morphometric analysis. (**a**) 3D cortical surfaces of in vivo, in silico, and in vitro models. Both left and right cortical surfaces are provided to present the left-right symmetry. (**b**) The quasi-conformal disk mapping with landmark matching of cortical surfaces on disk. Blue or red curves represent corresponding landmarks. Color represents shape index (SI). Similarity indices of each simulated and gel brain surfaces are presented in *Appendix 1—table 2*.

## Morphometric analysis

For the morphometric analysis, we developed a landmark-matching disk quasi-conformal parameterization method by extending and combining the methods in *Choi et al., 2015a*; *Choi and Lui, 2015b* and applied it to map the real, simulated, and gel brain surfaces onto the unit disk with major sulci aligned.

More specifically, we denote the real, simulated, and gel brain surfaces as $\mathcal{S}_1, \mathcal{S}_2, \mathcal{S}_3$, respectively, and note that $\mathcal{S}_1, \mathcal{S}_2, \mathcal{S}_3$ are all simply connected open surfaces and hence are topologically equivalent to the unit disk. We then started by applying the disk conformal parameterization method in *Choi et al., 2015a* to map $\mathcal{S}_1, \mathcal{S}_2, \mathcal{S}_3$ onto the unit disk, followed by the Möbius area correction scheme in *Choi et al., 2020* to further reduce the area distortion of the mappings. Denote the three-disk parameterization results as $\mathcal{D}_1, \mathcal{D}_2, \mathcal{D}_3$.

Next, we followed the idea in *Choi et al., 2015a* to compute landmark-aligned quasi-conformal maps between $\mathcal{D}_1, \mathcal{D}_2, \mathcal{D}_3$. More explicitly, given certain major sulci identified on each of the three brain surfaces $\mathcal{S}_1, \mathcal{S}_2, \mathcal{S}_3$, we computed two landmark-matching quasi-conformal maps $f_{13} : \mathcal{D}_1 \rightarrow \mathcal{D}_3$ and $f_{23} : \mathcal{D}_2 \rightarrow \mathcal{D}_3$ that deformed $\mathcal{D}_1$ and $\mathcal{D}_2$ to align the sulci on them with the corresponding sulci positions on $\mathcal{D}_3$. Consequently, we can visualize and compare the folding patterns of the three brain surfaces $\mathcal{S}_1, \mathcal{S}_2, \mathcal{S}_3$ by using the three disk parameterizations $f_{13}(\mathcal{D}_1), f_{23}(\mathcal{D}_2), \mathcal{D}_3$.

**Appendix 1—table 2.** Similarity index (main text, *Equation 4*) evaluated by rescaled mean curvature of simulated and gel brain surfaces with comparison to the real brain surfaces, calculated with different vector p-norm: $p = 1$, $p = 2$ and $p = \infty$.

| Similarity index | Simulated | | | Gel | | |
|---|---|---|---|---|---|---|
| $p$ | 1 | 2 | $\infty$ | 1 | 2 | $\infty$ |
| Ferret (left) | 0.8188 | 0.7541 | 1 | 0.7448 | 0.6746 | 1 |
| Ferret (right) | 0.7733 | 0.7005 | 1 | 0.7533 | 0.6887 | 1 |
| Macaque (left) | 0.6886 | 0.6039 | 1 | 0.7658 | 0.7086 | 1 |
| Macaque (right) | 0.6832 | 0.5994 | 1 | 0.7578 | 0.6989 | 1 |
| Human (left) | 0.6056 | 0.5090 | 1 | 0.7037 | 0.6540 | 1 |
| Human (right) | 0.5911 | 0.4909 | 1 | 0.7028 | 0.6525 | 1 |

## Diversity and evolution of cerebral folding in primates

For a surface-based (superficial) comparison of the brain morphologies among different primates from an evolutionary perspective, we take advantage of the fact that biologists have provided plenty of anatomical data on brain geometry, although a systematic study on how these cerebral folding patterns vary across different sizes, initial shapes, and foldedness for human and other primate species is still lacking. Among primates, brains vary enormously from roughly the size of a grape to the size of a grapefruit, and from nearly smooth to dramatically folded; of these, human brains are amongst the most folded and the largest (relative to body size). These variations in size and form make comparative neuroanatomy a rich resource for investigating common trends that transcend differences between species. We examined 10 primate species in order to cover a wide range of sizes and forms, as shown in *Appendix 1—figure 5*. Using our developed morphometric method, more investigations on the scaling law of their cortical thickness relative to the surface geometry, folding with respect to size (total surface area) and geometry (i.e. curvature, shape, and sulcal depth), and foldedness (gyrification) would be expected.

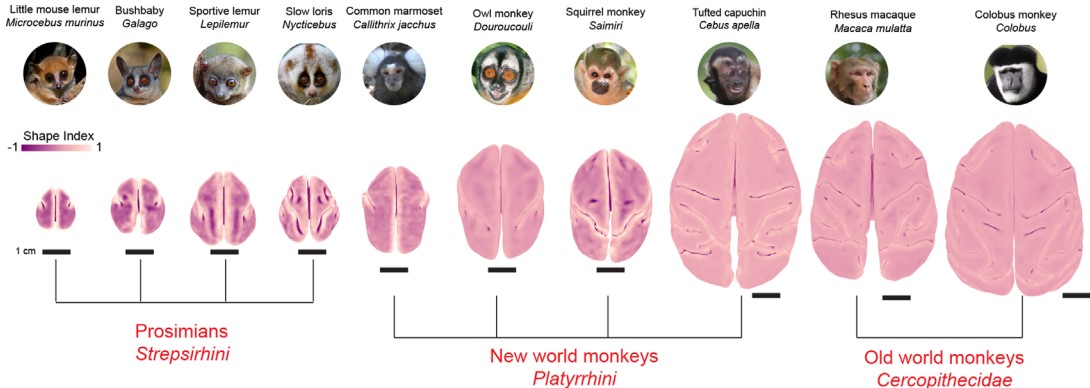

**Appendix 1—figure 5.** Comparison across 10 primate species. Each species is listed by its common name and scientific name, and accompanied by a picture. Scale bar: 1 cm. Color represents the shape index. Pictures are taken from Wikipedia.

