## [Editor Report · eLife Assessment]

This **important** study presents a cross-species and cross-disciplinary analysis of cortical folding. The authors use a combination of physical gel models, computational simulations, and morphometric analysis, extending prior work in human brain development to macaques and ferrets. The findings support the hypothesis that mechanical forces driven by differential growth can account for major aspects of gyrification. The evidence presented is overall strong and **convincingly** supports the central claims; the findings will be of broad interest in developmental neuroscience.

---

## [Referee Report · Reviewer #1 (Public review)]

The manuscript by Yin and colleagues addresses a long-standing question in the field of cortical morphogenesis, regarding factors that determine differential cortical folding across species and individuals with cortical malformations. The authors present work based on a computational model of cortical folding evaluated alongside a physical model that makes use of gel swelling to investigate the role of a two-layer model for cortical morphogenesis. The study assesses these models against empirically derived cortical surfaces based on MRI data from ferret, macaque monkey, and human brains.

The manuscript is clearly written and presented, and the experimental work (physical gel modeling as well as numerical simulations) and analyses (subsequent morphometric evaluations) are conducted at the highest methodological standards. It constitutes an exemplary use of interdisciplinary approaches for addressing the question of cortical morphogenesis by bringing together well-tuned computational modeling with physical gel models. In addition, the comparative approaches used in this paper establish a foundation for broad-ranging future lines of work that investigate the impact of perturbations or abnormalities during cortical development.

The cross-species approach taken in this study is a major strength of the work. However, correspondence across the two methodologies did not appear to be equally consistent in predicting brain folding across all three species. The results presented in Figures 4 (and Figures S3 & S4) show broad correspondence in shape index and major sulci landmarks across all three species. Nevertheless, the results presented for the human brain lack the same degree of clear correspondence for the gel model results as observed in the macaque and ferret. While this study clearly establishes a strong foundation for comparative cortical anatomy across species and the impact of perturbations on individual morphogenesis, further work that fine-tunes physical modeling of complex morphologies, such as that of the human cortex, may help to further understand the factors that determine cortical functionalization and pathologies.

---

## [Referee Report · Reviewer #2 (Public review)]

This manuscript explores the mechanisms underlying cerebral cortical folding using a combination of physical modelling, computational simulations, and geometric morphometrics. The authors extend their prior work on human brain development (Tallinen et al., 2014; 2016) to a comparative framework involving three mammalian species: ferrets (Carnivora), macaques (Old World monkeys), and humans (Hominoidea). By integrating swelling gel experiments with mathematical differential growth models, they simulate sulcification instability and recapitulate key features of brain folding across species. The authors make commendable use of publicly available datasets to construct 3D models of fetal and neonatal brain surfaces: fetal macaque (ref. [26]), newborn ferret (ref. [11]), and fetal human (ref. [22]).

Using a combination of physical models and numerical simulations, the authors compare the resulting folding morphologies to real brain surfaces using morphometric analysis. Their results show qualitative and quantitative concordance with observed cortical folding patterns, supporting the view that differential tangential growth of the cortex relative to the subcortical substrate is sufficient to account for much of the diversity in cortical folding. This is a very important point in our field, and can be used in the teaching of medical students.

Brain folding remains a topic of ongoing debate. While some regard it as a critical specialization linked to higher cognitive function, others consider it an epiphenomenon of expansion and constrained geometry. This divergence was evident in discussions during the Strüngmann Forum on cortical development (Silver et al., 2019). Though folding abnormalities are reliable indicators of disrupted neurodevelopmental processes (e.g., neurogenesis, migration), their relationship to functional architecture remains unclear. Recent evidence suggests that the absolute number of neurons varies significantly with position-sulcus versus gyrus-with potential implications for local processing capacity (e.g., https://doi.org/10.1002/cne.25626). The field is thus in need of comparative, mechanistic studies like the present one.

This paper offers an elegant and timely contribution by combining gel-based morphogenesis, numerical modelling, and morphometric analysis to examine cortical folding across species. The experimental design - constructing two-layer PDMS models from 3D MRI data and immersing them in organic solvents to induce differential swelling - is well-established in prior literature. The authors further complement this with a continuum mechanics model simulating folding as a result of differential growth, as well as a comparative analysis of surface morphologies derived from in vivo, in vitro, and in silico brains.

Conclusion:

This is a well-executed and creative study that integrates diverse methodologies to address a longstanding question in developmental neurobiology. While a few aspects-such as regional folding peculiarities, sensitivity to initial conditions, and available human data-could be further elaborated, they do not detract from the overall quality and novelty of the work. I enthusiastically support this paper and believe that it will be of broad interest to the neuroscience, biomechanics, and developmental biology communities.

[Editor's note: The reviewers were satisfied with the authors' response. The eLife Assessment was slightly updated to reflect the author's response.]

---

## [Author Response]

The following is the authors’ response to the original reviews

**Public Reviews:**

**Reviewer #1 (Public review):**
The manuscript by Yin and colleagues addresses a long-standing question in the field of cortical morphogenesis, regarding factors that determine differential cortical folding across species and individuals with cortical malformations. The authors present work based on a computational model of cortical folding evaluated alongside a physical model that makes use of gel swelling to investigate the role of a two-layer model for cortical morphogenesis. The study assesses these models against empirically derived cortical surfaces based on MRI data from ferret, macaque monkey, and human brains.The manuscript is clearly written and presented, and the experimental work (physical gel modeling as well as numerical simulations) and analyses (subsequent morphometric evaluations) are conducted at the highest methodological standards. It constitutes an exemplary use of interdisciplinary approaches for addressing the question of cortical morphogenesis by bringing together well-tuned computational modeling with physical gel models. In addition, the comparative approaches used in this paper establish a foundation for broad-ranging future lines of work that investigate the impact of perturbations or abnormalities during cortical development.The cross-species approach taken in this study is a major strength of the work. However, correspondence across the two methodologies did not appear to be equally consistent in predicting brain folding across all three species. The results presented in Figures 4 (and Figures S3 and S4) show broad correspondence in shape index and major sulci landmarks across all three species. Nevertheless, the results presented for the human brain lack the same degree of clear correspondence for the gel model results as observed in the macaque and ferret. While this study clearly establishes a strong foundation for comparative cortical anatomy across species and the impact of perturbations on individual morphogenesis, further work that fine-tunes physical modeling of complex morphologies, such as that of the human cortex, may help to further understand the factors that determine cortical functionalization and pathologies.

We thank the reviewer for positive opinions and helpful comments. Yes, the physical gel model of the human brain has a lower similarity index with the real brain. There are several reasons.

First, the highly convoluted human cortex has a few major folds (primary sulci) and a very large number of minor folds associated with secondary or tertiary sulci (on scales of order comparable to the cortical thickness), relative to the ferret and macaque cerebral cortex. In our gel model, the exact shapes, positions, and orientations of these minor folds are stochastic, which makes it hard to have a very high similarity index of the gel models when compared with the brain of a single individual.

Second, in real human brains, these minor folds evolve dynamically with age and show differences among individuals. In experiments with the gel brain, multiscale folds form and eventually disappear as the swelling progresses through the thickness. Our physical model results are snapshots during this dynamical process, which makes it hard to have a concrete one-to-one correspondence between the instantaneous shapes of the swelling gel and the growing human brain.

Third, the growth of the brain cortex is inhomogeneous in space and varying with time, whereas, in the gel model, swelling is relatively homogeneous.

We agree that further systematic work, based on our proposed methods, with more fine-tuned gel geometries and properties, might provide a deeper understanding of the relations between brain geometry, and growth-induced folds and their functionalization and pathologies. Further analysis of cortical pathologies using computational and physical gel models can be found in our companion paper (Choi et al., 2025), also published in eLife:

G. P. T. Choi, C. Liu, S. Yin, G. Séjourné, R. S. Smith, C. A. Walsh, L. Mahadevan, Biophysical basis for brain folding and misfolding patterns in ferrets and humans. eLife, 14, RP107141, 2025. doi:10.7554/eLife.107141

**Reviewer# 2 (Public review):**
This manuscript explores the mechanisms underlying cerebral cortical folding using a combination of physical modelling, computational simulations, and geometric morphometrics. The authors extend their prior work on human brain development (Tallinen et al., 2014; 2016) to a comparative framework involving three mammalian species: ferrets (Carnivora), macaques (Old World monkeys), and humans (Hominoidea). By integrating swelling gel experiments with mathematical differential growth models, they simulate sulcification instability and recapitulate key features of brain folding across species. The authors make commendable use of publicly available datasets to construct 3D models of fetal and neonatal brain surfaces: fetal macaque (ref. [26]), newborn ferret (ref. [11]), and fetal human (ref. [22]).Using a combination of physical models and numerical simulations, the authors compare the resulting folding morphologies to real brain surfaces using morphometric analysis. Their results show qualitative and quantitative concordance with observed cortical folding patterns, supporting the view that differential tangential growth of the cortex relative to the subcortical substrate is sufficient to account for much of the diversity in cortical folding. This is a very important point in our field, and can be used in the teaching of medical students.Brain folding remains a topic of ongoing debate. While some regard it as a critical specialization linked to higher cognitive function, others consider it an epiphenomenon of expansion and constrained geometry. This divergence was evident in discussions during the Strungmann Forum on cortical development (Silver¨ et al., 2019). Though folding abnormalities are reliable indicators of disrupted neurodevelopmental processes (e.g., neurogenesis, migration), their relationship to functional architecture remains unclear. Recent evidence suggests that the absolute number of neurons varies significantly with position-sulcus versus gyrus-with potential implications for local processing capacity (e.g., https://doi.org/10.1002/cne.25626). The field is thus in need of comparative, mechanistic studies like the present one.This paper offers an elegant and timely contribution by combining gel-based morphogenesis, numerical modelling, and morphometric analysis to examine cortical folding across species. The experimental design - constructing two-layer PDMS models from 3D MRI data and immersing them in organic solvents to induce differential swelling - is well-established in prior literature. The authors further complement this with a continuum mechanics model simulating folding as a result of differential growth, as well as a comparative analysis of surface morphologies derived from in vivo, in vitro, and in silico brains.

We thank the reviewer for the very positive comments.

I offer a few suggestions here for clarification and further exploration:Major Comments(1) Choice of Developmental Stages and Initial ConditionsThe authors should provide a clearer justification for the specific developmental stages chosen (e.g., G85 for macaque, GW23 for human). How sensitive are the resulting folding patterns to the initial surface geometry of the gel models? Given that folding is a nonlinear process, early geometric perturbations may propagate into divergent morphologies. Exploring this sensitivity-either through simulations or reference to prior work-would enhance the robustness of the findings.

The initial geometry is one of the important factors that decides the final folding pattern. The smooth brain in the early developmental stage shows a broad consistency across individuals, and we expect the main folds to form similarly across species and individuals.

Generally, we choose the initial geometry when the brain cortex is still relatively smooth. For the human, this corresponds approximately to GW23, as the major folds such as the Rolandic fissure (central sulcus), arise during this developmental stage. For the macaque brain, we chose developmental stage G85, primarily because of the availability of the dataset corresponding to this time, which also corresponds to the least folded.

We expect that large-scale folding patterns are strongly sensitive to the initial geometry but fine-scale features are not. Since our goal is to explain the large-scale features, we expect sensitivity to the initial shape.

Below are some references of other researchers that are consistent with this idea. Figure 4 from Wang et al. shows some images of simulations obtained by perturbing the geometry of a sphere to an ellipsoid. We see that the growth-induced folds mostly maintain their width (wavelength), but change their orientations.

Reference:

Wang, X., Lefévre, J., Bohi, A., Harrach, M.A., Dinomais, M. and Rousseau, F., 2021. The influence of biophysical parameters in a biomechanical model of cortical folding patterns. Scientific Reports, 11(1), p.7686.

Related results from the same group show that slight perturbations of brain geometry, cause these folds also tend to change their orientations but not width/wavelength (Bohi et al., 2019).

Reference:

Bohi, A., Wang, X., Harrach, M., Dinomais, M., Rousseau, F. and Lefévre, J., 2019, July. Global perturbation of initial geometry in a biomechanical model of cortical morphogenesis. In 2019 41st Annual International Conference of the IEEE Engineering in Medicine and Biology Society (EMBC) (pp. 442-445). IEEE.

Finally, a systematic discussion of the role of perturbations on the initial geometries and physical properties can be seen in our work on understanding a different system, gut morphogenesis (Gill et al., 2024).

We have added the discussion about geometric sensitivity in the section Methods-Numerical Simulations:

“Small perturbations on initial geometry would affect minor folds, but the main features of major folds, such as orientations, width, and depth, are expected to be conserved across individuals [49, 50]. For simplicity, we do not perturb the fetal brain geometry obtained from datasets.”

(2) Parameter Space and Breakdown PointsThe numerical model assumes homogeneous growth profiles and simplifies several aspects of cortical mechanics. Parameters such as cortical thickness, modulus ratios, and growth ratios are described in Table II. It would be informative to discuss the range of parameter values for which the model remains valid, and under what conditions the physical and computational models diverge. This would help delineate the boundaries of the current modelling framework and indicate directions for refinement.

Exploring the valid parameter space is a key problem. We have tested a series of growth parameters and will state them explicitly in our revision. In the current version, we chose the ones that yield a relatively high similarity index to the animal brains. More generally, folding patterns are largely regulated by geometry as well as physical parameters, such as cortical thickness, modulus ratios, growth ratios, and inhomogeneity. In our previous work on a different system, gut morphogenesis, where similar folding patterns are seen, we have explored these features (Gill et al., 2024).

Reference:

Gill, H.K., Yin, S., Nerurkar, N.L., Lawlor, J.C., Lee, C., Huycke, T.R., Mahadevan, L. and Tabin, C.J., 2024. Hox gene activity directs physical forces to differentially shape chick small and large intestinal epithelia. Developmental Cell, 59(21), pp.2834-2849.

(3) Neglected Regional Features: The Occipital Pole of the MacaqueOne conspicuous omission is the lack of attention to the occipital pole of the macaque, which is known to remain smooth even at later gestational stages and has an unusually high neuronal density (2.5× higher than adjacent cortex). This feature is not reproduced in the gel or numerical models, nor is it discussed. Acknowledging this discrepancy-and speculating on possible developmental or mechanical explanationswould add depth to the comparative analysis. The authors may wish to include this as a limitation or a target for future work.

Yes, we have added that the omission of the Occipital Pole of the macaque is one of our paper’s limitations. Our main aim in this paper is to explore the formation of large-scale folds, so the smooth region is not discussed. But future work could include this to make the model more complete.

The main text has been modified in Methods, Numerical simulations:

“To focus on fold formation, we did not discuss the relatively smooth region, such as the Occipital Pole of the macaque.”

and also in the caption of Figure 4: “... The occipital pole region of macaque brains remains smooth in real and simulated brains.”

(4) Spatio-Temporal Growth Rates and Available Human DataThe authors note that accurate, species-specific spatio-temporal growth data are lacking, limiting the ability to model inhomogeneous cortical expansion. While this may be true for ferret and macaque, there are high-quality datasets available for human fetal development, now extended through ultrasound imaging (e.g., https://doi.org/10.1038/s41586-023-06630-3). Incorporating or at least referencing such data could improve the fidelity of the human model and expand the applicability of the approach to clinical or pathological scenarios.

We thank the reviewer for pointing out the very useful datasets that exist for the exploration of inhomogeneous growth driven folding patterns. We have referred to this paper to provide suggestions for further work in exploring the role of growth inhomogeneities.

We have referred to this high-quality dataset in our main text, Discussion:

“...the effect of inhomogeneous growth needs to be further investigated by incorporating regional growth of the gray and white matter not only in human brains [29, 31] based on public datasets [45], but also in other species.”

A few works have tried to incorporate inhomogeneous growth in simulating human brain folding by separating the central sulcus area into several lobes (e.g., lobe parcellation method, Wang, PhD Thesis, 2021). Since our goal in this paper is to explain the large-scale features of folding in a minimal setting, we have kept our model simple and show that it is still capable of capturing the main features of folding in a range of mammalian brains.

Reference:

Xiaoyu Wang. Modélisation et caractérisation du plissement cortical. Signal and Image Processing. Ecole nationale superieure Mines-Télécom Atlantique, 2021. English. 〈NNT : 2021IMTA0248〉.

(5) Future Applications: The Inverse Problem and Fossil BrainsThe authors suggest that their morphometric framework could be extended to solve the inverse growth problem-reconstructing fetal geometries from adult brains. This speculative but intriguing direction has implications for evolutionary neuroscience, particularly the interpretation of fossil endocasts. Although beyond the scope of this paper, I encourage the authors to elaborate briefly on how such a framework might be practically implemented and validated.

For the inverse problem, we could use the following strategies:

a. Perform systematic simulations using different geometries and physical parameters to obtain the variation in morphologies as a function of parameters.

b. Using either supervised training or unsupervised training (physics-informed neural networks, PINNs) to learn these characteristic morphologies and classify their dependence on the parameters using neural networks. These can then be trained to determine the possible range of geometrical and physical parameters that yield buckled patterns seen in the systematic simulations.

c. Reconstruct the 3D surface from fossil endocasts. Using the well-trained neural network, it should be possible to predict the initial shape of the smooth brain cortex, growth profile, and stiffness ratio of the gray and white matter.

As an example in this direction, supervised neural networks have been used recently to solve the forward problem to predict the buckling pattern of a growing two-layer system (Chavoshnejad et al., 2023). The inverse problem can then be solved using machine-learning methods when the training datasets are the folded shape, which are then used to predict the initial geometry and physical properties.

Reference:

Chavoshnejad, P., Chen, L., Yu, X., Hou, J., Filla, N., Zhu, D., Liu, T., Li, G., Razavi, M.J. and Wang, X., 2023. An integrated finite element method and machine learning algorithm for brain morphology prediction. Cerebral Cortex, 33(15), pp.9354-9366.

ConclusionThis is a well-executed and creative study that integrates diverse methodologies to address a longstanding question in developmental neurobiology. While a few aspects-such as regional folding peculiarities, sensitivity to initial conditions, and available human data-could be further elaborated, they do not detract from the overall quality and novelty of the work. I enthusiastically support this paper and believe that it will be of broad interest to the neuroscience, biomechanics, and developmental biology communities.Note: The paper mentions a companion paper [reference 11] that explores the cellular and anatomical changes in the ferret cortex. I did not have access to this manuscript, but judging from the title, this paper might further strengthen the conclusions.

The companion paper (Choi et al., 2025) has also been submitted to eLife and can be found here:

G. P. T. Choi, C. Liu, S. Yin, G. Séjourné, R. S. Smith, C. A. Walsh, L. Mahadevan, Biophysical basis for brain folding and misfolding patterns in ferrets and humans. eLife, 14, RP107141, 2025. doi:10.7554/eLife.107141

**Recommendations for the authors:**

**Reviewer #1 (Recommendations for the authors):**
This study was conducted and presented to the highest methodological standards. It is clearly written, and the results are thoroughly presented in the main manuscript and supplementary materials. Nevertheless, I would present the following minor points and comments for consideration by the authors prior to finalizing their work:

We thank the reviewer for positive opinions and helpful comments.

(1) Where did the MRI-based cortical surface data come from? Specifically, it would be helpful to include more information regarding whether the surfaces were reconstructed based on individual- or group-level data. It appears the surfaces were group-level, and, if so, accounting for individual-level cortical folding may be a fruitful direction for future work.

The surface data come from public database, which are stated in the Methods Section. “We used a publicly available database for all our 3d reconstructions: fetal macaque brain surfaces are obtained from Liu et al. (2020); newborn ferret brain surfaces are obtained from Choi et al. (2025); and fetal human brain surfaces are obtained from Tallinen et al. (2016).”

These surfaces are reconstructed based on group-level data. Specifically, the macaque atlas images are constructed for brains at gestational ages of 85 days (G85, *N* = 18_,_ 9 females), 110 days (G110, *N* = 10_,_ 7 females) and 135 days (G135, *N* = 16_,_ 7 females). And yes, future work may focus on individual-level cortical folding, and we expect that more specific results could be found.

(2) One methodological approach for assessing consistency of cortical folding within species might be an evaluation of cross-hemispheric symmetry. I would find this particularly interesting with respect to the gel models, as it could complement the quantification of variation with respect to the computationally derived and real surfaces.

Yes, the cross-hemispheric symmetry comparison can be done by our morphometric analysis method. We have added the results of ferret brain’s left-right symmetry for gel models, simulations, and real surfaces in the supplementary material. A typical conformal mapping figure and the similarity index table are shown here.

(3) Was there a specific reason to reorder the histogram plots in Figure 4c to macaque, ferret, human rather than to maintain the order presented in Figure 4a/b of ferret, macaque, human? I appreciate that this is a minor concern, and all subplots are indeed properly titled, but consistent order may improve clarity.

We have reordered the histogram plots to make all the figure orders consistent.

**Reviewer #2 (Recommendations for the authors):**
(1) Please consider revising the caption of Figure 1 (or equivalent figures) to explicitly state whether features such as the macaque occipital flatness were reproduced or not.

We thank the reviewer for pointing out the macaque occipital flatness.

**Author response table 1. sa3table1:** Left-right similarity index evaluated by comparing the shape index of ferret brains, calculated with vector P-NORM *p*=2,.

real	gel	simu
0.5754	0.7014	0.5868

**Author response image 1. sa3fig1:** Left-right similarity index of ferret brains.

Occipital Pole of the macaque remains relatively smooth in both real brains and computational models. But our main aim in this paper is to explore the large-scale folds formation, so the smooth region is not discussed in depth. But future work could include this to make the model more complete.

(2) Some figures could benefit from clearer labelling to distinguish between in vivo, in vitro, and in silico results.

We have supplemented some texts in panels to make the labelling clearer.

(3) The manuscript would benefit from a short paragraph in the Discussion reflecting on how future incorporation of regional heterogeneities might improve model fidelity.

We have added a sentence in the Discussion Section about improving the model fidelity by considering regional heterogeneities.

“Future more accurate models incorporating spatio-temporal inhomogeneous growth profiles and mechanical properties, such as varying stiffness, would make the folding pattern closer to the real cortical folding. This relies on more in vivo measurements of the brain’s physical properties and cortical expansion.”

(4) Suggestions for improved or additional experiments, data, or analyses.(5) Clarify and justify the selection of developmental stages: The authors should explain why particular gestational stages (e.g., G85 for macaque, GW23 for human) were chosen as starting points for the physical and computational models. A discussion of how sensitive the folding patterns are to the initial geometry would help assess the robustness of the model. If feasible, a brief sensitivity analysis-varying initial age or surface geometry-would strengthen the conclusions.

The initial geometry is one of the important factors that decides the final folding pattern. The smooth brain in the early developmental stage shows a broad consistency across individuals, and we expect the main folds to form similarly across species and individuals.

Generally, we choose the initial geometry when the brain cortex is still relatively smooth. For the human, this corresponds approximately to GW23, as the major folds such as the Rolandic fissure (central sulcus), arise during this developmental stage. For the macaque brain, we chose developmental stage G85, primarily because of the availability of the dataset corresponding to this time, which also corresponds to the least folded.

We expect that large-scale folding patterns are strongly sensitive to the initial geometry but fine-scale features are not. Since our goal is to explain the large-scale features, we expect sensitivity to the initial shape.

We have added the discussion about geometric sensitivity in the section Methods-Numerical Simulations: “Small perturbations on initial geometry would affect minor folds, but the main features of major folds, such as orientations, width, and depth, are expected to be conserved across individuals [49, 50]. For simplicity, we do not perturb the fetal brain geometry obtained from datasets.”

(6) Explore parameter boundaries more explicitly: The paper would benefit from a clearer account of the ranges of mechanical and geometric parameters (e.g., growth ratios, cortical thickness) for which the model holds. Are there specific conditions under which the physical and numerical models diverge? Identifying breakdown points would help readers understand the model’s limitations and applicability.

Exploring the valid parameter space is a key problem. We have tested a series of growth parameters and will state them explicitly in our revision. In the current version, we chose the ones that yield a relatively high similarity index to the animal brains. More generally, folding patterns are largely regulated by geometry as well as physical parameters, such as cortical thickness, modulus ratios, and growth ratios and inhomogeneity. In our previous work on a different system, gut morphogenesis, where similar folding patterns are seen, we have explored these features (Gill et al., 2024).

(7) Address species-specific cortical peculiarities: A striking omission is the flat occipital pole of the macaque, which is not reproduced in the physical or computational models. Given its known anatomical and cellular distinctiveness, this discrepancy warrants discussion. Even if not explored experimentally, the authors could speculate on what developmental or mechanical conditions would be needed to reproduce such regional smoothness.

Please refer to our answer to the public reviewer 2, question (3). From our results, the formation of smooth Occipital Pole might indicate that the spatio-temporal growth rate of gray and white matter are consistent in this region, such that there’s no much differential growth.

(8) Consider integration of available human growth data: While the authors note the lack of spatiotemporal growth data across species, such datasets exist for human fetal brain development, including those from MRI and ultrasound studies (e.g., Nature 2023). Incorporating these into the human model-or at least discussing their implications-would enhance biological relevance.

Yes, some datasets for fetal human brains have provided very comprehensive measurements on brain shapes at many developmental stages. This can surely be implemented in our current model by calculating the spatio-temporal growth rate from regional cortical shapes at different stages.

(9) Recommendations for improving the writing and presentation:a) The manuscript is generally well-written, but certain sections would benefit from more explicit linksbetween the biological phenomena and the modeling framework. For instance, the Introduction and Discussion could more clearly articulate how mechanical principles interface with genetic or cellular processes, especially in the context of evolution and developmental variation.

We have briefly discussed the gene-regulated cellular process and the induced changes of mechanical properties and growth rules in SI, table S1. In the main text, to be clearer, we have added a sentence:

“Many malformations are related to gene-regulated abnormal cellular processes and mechanical properties, which are discussed in SI”

b) The Discussion could better acknowledge limitations and future directions, including regional dif-ferences in folding, inter-individual variability, and the model’s assumptions of homogeneous material properties and growth.

In the discussion section, we have pointed out four main limitations and open directions based on our current model, including the discussion on spatiotemporal growth and property. To be more complete, we have supplemented other limitations on the regional differences in folding and the interindividual variability. In the main text, we added the following sentence:

“In addition to the homogeneity assumption, we have not investigated the inter-individual variability and regional differences in folding. More accurate and specific work is expected to focus on these directions.”

c) The authors briefly mention the potential for addressing the inverse growth problem. Expanding this idea in a short paragraph - perhaps with hypothetical applications to fossil brain reconstructions-would broaden the paper’s appeal to evolutionary neuroscientists.

We have stated general steps in the response to public reviewer 2, question (5).

(10) Minor corrections to the text and figures:a) Figures:Label figures more clearly to distinguish between in vivo, in vitro, and in silico brain representations.– Ensure that the occipital pole of the macaque is visible or annotated, especially if it lacks the expected smoothness.Add scale bars where missing for clarity in morphometric comparisons.

We thank the reviewer for suggestions to improve the readability of our manuscript.

The in vivo (real), in vitro (gel), and in silico (simulated) results are both distinguished by their labels and different color scheme: gray-white for real brain, pink-white for gel model, and blue-white for simulations, respectively.

The occipital pole of the macaque brain remains relatively smooth in our computational model but notin our physical gel model. We have clarified this in the main text: “To focus on fold formation, we did not discuss the relatively smooth region, such as the Occipital Pole of the macaque.”

All the brain models are rescaled to the same size, where the distance between the anterior-most pointof the frontal lobe and the posterior-most point of the occipital lobe is two units.

b) Text:Consider revising figure captions to explicitly mention whether specific regional features (e.g., flatoccipital pole) were observed or absent in models.In Table II (and relevant text), ensure parameter definitions are consistent and explained clearly for across-disciplinary audience.Add citations to recent human fetal growth imaging work (e.g., ultrasound-based studies) to support claims about available data.

We have added some descriptions of the characters of the folding pattern in the caption of Figure 4,including major folds and smooth regions.

“Three or four major folds of each brain model are highlighted and served as landmarks. The occipital pole region of macaque brains remains smooth in real and simulated brains.”

We have clarified the definition of growth ratio *g*Msub>g/*g*_w_ and stiffness ratio *µ*_g_/*µ*_w_ between gray matter and white matter, and the normalized cortical thickness *h/L* in Table 2.

We have referred to a high-quality dataset of fetal brain imaging work, the ultrasound-imaging method(Namburete et al. 2023), in our main text, Discussion:

“...the effect of inhomogeneous growth needs to be further investigated by incorporating regional growth of the gray and white matter not only in human brains [29, 31] based on public datasets [45], but also in other species.”